# An Indispensable Requirement for Medical Dosage Calculation: Basic Mathematical Skills of Baccalaureate Nursing Students

**DOI:** 10.3390/nursrep15050150

**Published:** 2025-04-30

**Authors:** Belal Mahmoud Hijji

**Affiliations:** Department of Nursing, College of Applied Medical Sciences, Prince Sattam bin Abdulaziz University, Wadi ad-Dawasser 11991, Saudi Arabia; b.higah@psau.edu.sa; Tel.: +966-550351102

**Keywords:** mathematical skills, nursing education, patient safety, nursing

## Abstract

**Background/Objectives**: While drug dosage calculation is vital in nursing, research indicates nursing students often struggle with necessary mathematics competencies, a gap not previously explored in the Arab world. This study assessed the basic mathematical skills of baccalaureate nursing students in a branch of a Saudi Arabian public university and compared the findings with studies conducted in other countries, which have consistently reported better performance. By highlighting these disparities, this study underscored the need for global educational reforms to ensure safe nursing practices. **Methods**: This was a cross-sectional study. Three hundred and thirty students were invited; consenting students completed a mathematics experts-validated 45-question test covering four key areas: numbers and operations, data interpretation, measurement, and algebraic applications. Descriptive and inferential statistics were applied. The Mann–Whitney U test was used to detect differences in scores based on gender. An independent-samples Kruskal–Wallis test was conducted to compare the three student groups simultaneously. As this test was statistically significant, post hoc pairwise comparisons were performed to assess differences in scores between the first and second, first and third, and second and third levels of study. **Results**: A response rate of 40.6% was achieved. Scores ranged from 3 to 58 (median: 18, 27%), with only 2% passing (≥60%). Significant differences in scores were found between genders (*p* = 0.037) and across study levels (*p* = 0.002). Overall, 25 (56%) items were difficult, while 20 (44%) were moderately difficult. **Conclusions**: The low median score underscored a critical need for interventions to improve mathematical competencies in nursing students, affecting medication safety in healthcare systems.

## 1. Introduction

Quality education is a powerful driver of a nation’s prosperity and progress [1]. Within this context, universities play a crucial role in producing safe and accountable nurses [2] who are equipped to provide effective patient care. A critical component of nursing education and clinical practice is drug dosage calculation [3], as nurses are primarily responsible for medication administration [4]. To fulfill this responsibility, mastering basic mathematical skills is an essential prerequisite [3] and a fundamental aspect of professional nursing practice [5]. Without these skills, nursing students would be unable to perform their duties safely and effectively [6].

However, research shows that many nursing students struggle with basic mathematical calculations [3], an issue that often persists after graduation [7] and significantly contributes to medication administration errors (MAEs) [8]. One study suggests that tracking nursing graduates longitudinally could provide valuable insights into whether fewer medication errors occur over time as they gain more clinical experience [9].

These errors do not occur in isolation but have underlying causes. As students transition to university, they may gradually forget the skills required for solving mathematical problems [10]. Additionally, a lack of both conceptual and procedural knowledge necessary for solving mathematical problems [11,12] further compounds the issue. Research shows that most mathematical errors among nursing students stem from conceptual misunderstandings [13], though procedural errors also play a significant role [14]. These challenges not only affect clinical practice but also highlight deeper issues in conceptual and procedural knowledge acquisition.

The consequences of these preventable MAEs vary in severity and can impact patients, families, and the healthcare system. Most critically, MAEs can directly contribute to patient morbidity and mortality [15]. In Ireland, incorrect medication dosing accounted for 90% of all MAEs [16], and, in other countries with advanced educational systems, incorrect dosing was linked to significant patient harm and fatalities, such as 23 (14%) of 167 deaths in Australia [17] and 13 (48%) of 27 deaths in Norway [18]. Additionally, the economic burden associated with MAEs is substantial [16,19]. These errors also negatively impact nurses, leading to psychological trauma, a loss of confidence in their clinical abilities [20], and even job loss [21].

Given the importance of basic mathematical skills for accurate medical dosage calculation, this area has been a focus of research for decades [22,23], continuing into more recent studies [3,24,25,26]. Although various studies highlight numeracy challenges across healthcare education worldwide, particularly in the UK, Turkey, and Italy, limited research has focused on Arab countries, including Saudi Arabia, leaving a critical gap in understanding regional trends. This gap is particularly significant given the region’s large population of 462,523,909 [27]. Additionally, the relationship between basic mathematical skills, gender, and level of study remains underexplored.

Addressing this gap requires directly assessing nursing students’ basic mathematical skills, as recommended [28]. This necessity is further emphasized by the standards set for university non-medical programs by the United Kingdom’s Nursing and Midwifery Council [29].

## 2. Materials and Methods

### 2.1. Study Aims and Objectives

This study aimed to assess the basic mathematical skills of all undergraduate nursing students at a branch of a public Saudi Arabian university and to identify associated factors.

The specific objectives were the following:Assessing nursing students’ skills in solving problems related to algebraic applications, data interpretation, numbers and operations, and measurement.Determining whether statistically significant differences in performance exist between male and female students.Determining whether statistically significant differences in performance exist among students based on their level of study.Computing the difficulty index (*p*) of the test items.

### 2.2. Study Design

This study adopted a descriptive and cross-sectional design. While this design offers valuable insights into students’ mathematical competencies, it limits causal interpretation of the underlying factors.

### 2.3. Setting and Sample

This study was conducted at a university branch with separate male and female campuses located near each other in the Ar-Riyadh region. Three hundred and thirty (female = 111; male = 219) students, comprising the target population from all levels of study, were invited to participate in this study. A convenience sampling method was used, where any student who expressed interest in participation was included. No determination of sample size was made because the aim was to recruit as many students as possible from the target population. Recruitment of the sample in the male campus commenced in February 2022 and was completed early in February 2023. In the female campus, sample recruitment commenced in December 2022 and completed in September 2023.

### 2.4. Ethical Considerations

Ethical approval for this study was granted by the university where this study was conducted. Administrative approval was also obtained from the Dean of the College of Applied Medical Sciences. The researcher and a female nursing colleague approached male and female students, respectively, providing each with a participant information sheet about this study and inviting voluntary participation. All the students were assured of confidentiality and anonymity; those who agreed to participate signed an informed consent form. They were assured that their participation was voluntary and that they could withdraw at any time without any penalty.

### 2.5. Data Collection Tool

Since there was no standardized test used across previous studies, the initial aim of this study was to collaborate with the American Assessment Technologies Institute (ATI) to administer the Test of Essential Academic Skills (TEAS) online. However, as this was not feasible, the researcher and the colleague constructed a test based on the mathematics portion of the TEAS. The original test comprised 50 questions, with 16 requiring calculations and written responses, and 34 multiple-choice questions. However, following the content validity review (described below), five items were eliminated, resulting in a final test of 45 questions—16 requiring calculations and written responses, and 29 multiple-choice questions. The test was designed to be completed without the use of a calculator to ensure that results reflected students’ fundamental mathematical skills.

The test was modeled after TEAS mathematics portion to ensure comparability with international standards, reflecting essential competencies required for nursing students globally. This portion focuses on four key areas: numbers and operations, data interpretation, measurement, and algebraic applications [30].

The number of questions allocated to each area was as follows:

Numbers and operations: 17.

Data interpretation: 5.

Measurement: 13.

Algebraic applications: 10.

These four areas have ten components that the test addressed:

Component 1: converting between decimals, fractions, and percentages (*n* = 4).

Component 2: adding and dividing fractions and mixed numbers (*n* = 4).

Component 3: solving problems containing ratios, proportions, and rate of change (*n* = 4).

Component 4: converting between Roman and Arabic numbers (*n* = 4).

Component 5: solving equations with one unknown variable (*n* = 4).

Component 6: solving equations and inequalities with or without absolute values (*n* = 6).

Component 7: organizing and interpreting data from tables, graphs, and charts (*n* = 5).

Component 8: converting measurements (*n* = 7).

Component 9: estimating metric quantities (*n* = 3).

Component 10: measuring the dimensions, weight, and volume of objects (*n* = 4).

### 2.6. Content Validity and Scoring of the Test

A panel of five mathematics experts reviewed the original 50 items for content validity using a four-point rating scale: 1 = not relevant; 2 = needs revision; 3 = relevant but requires minor changes; and 4 = very relevant and succinct [31]. A content validity index (CVI) of 0.80 for individual items is considered acceptable [31]. The validation process revealed that 36 items achieved a CVI of 1.00, and 9 items had a CVI of 0.80, meaning they were considered content-valid. None of the items were rated 1 or 2 by all experts. However, five items received lower ratings, with CVIs of 0.60, 0.40, and 0.20 (two items each for 0.60 and 0.40, and one item for 0.20). Following Polit and Beck [32], the overall CVI of the test was calculated and found to be 0.91, which is acceptable [33]. Based on the experts’ feedback, the five items with a CVI of ≤0.60 were eliminated, which raised the overall CVI of the test to 0.96. Additionally, the expert comments were used to revise and improve the remaining items as recommended [31].

Following the content validation, four of the five experts provided input on the point allocation for each test item. There was significant variation in their opinions, with little consensus on most items. To resolve this, we assigned a final score to each item by calculating the mean of the experts’ suggested points (total points divided by the number of experts), which is standard practice when expert opinions differ substantially [34]. Ultimately, 22 items were assigned 22 points, and the remaining 23 items were worth 44 points.

### 2.7. Data Collection Process

On the date of data collection, students were given the test papers, where, at the top of the first page, they were advised that using a calculator was not allowed, to maintain the integrity of the test and produce reliable results. Each page of the test had enough space for the students to use. The students were given as much time as they needed to complete the test.

To preserve anonymity, the students were instructed not to include their names or university numbers on the test. They were also advised that the researcher and his colleague were ready to translate any word or phrase into Arabic if needed. Other than this, no translation assistance protocol was used. The students completed the test under the supervision of the researcher and his colleague to ensure that they could not communicate with each other, as recommended [35], and to ensure that they did not miss any question. If students asked, “What should I do if I cannot answer a question?”, they were advised to skip that question and move on to the next one. In the male campus, data collection commenced late in December 2022 and was completed late in February 2023. In the female campus, data collection commenced in December 2022 and completed in September 2023.

### 2.8. Item Analysis

Item analysis is a statistical evaluation of students’ responses to a test [36]. Item difficulty index (*p* value) is a key aspect of item analysis [37], which is the proportion of students who answered an item correctly [38]. To determine the difficulty level, students’ answers were analyzed, and the *p* value for each item was calculated. An item was classified as easy (*p* > 0.80), moderately difficult (*p* between 0.30 and 0.80), or difficult (*p* < 0.30) [39].

### 2.9. Data Analysis

The data were coded and entered into IBM SPSS^®^ version 27. Descriptive and inferential statistics were used to analyze the data. Descriptive statistics such as frequency ratings and percentages were calculated for nominal-level data (gender, level of study, and students’ responses to knowledge test items, which were coded as correct or incorrect). Additionally, descriptive statistics and the Kolmogorov–Smirnov test were used to assess the normality of score distributions. The Mann–Whitney U test was used to detect differences in scores based on gender (male vs. female). An independent-samples Kruskal–Wallis test was conducted to compare the three student groups simultaneously. As this test was statistically significant, post hoc pairwise comparisons were performed to assess differences in scores between the first and second, first and third, and second and third levels of study. The significance values were adjusted using the Bonferroni correction, as reported by the statistical software.

A passing score in Saudi Arabian universities is 60% (40 out of 66). As such, any student who achieved this score was considered successful.

## 3. Results

### 3.1. Demographic Characteristics

Out of the 330 students invited to participate, a total of 134 students completed the test, resulting in a 40.6% response rate. Of these, 96 (72%) were male and 38 (28%) female. The number and percentage of students in the first year, second year, and third year/seventh level of fourth year were 46 (34.4%), 58 (43.3%), and 30 (22.4%), respectively.

### 3.2. Results of the Test

The nursing students’ knowledge scores were generally very low, ranging from 3 to 58 out of 66 (5% to 88%). Only a few students (*n* = 3, 2%) passed the test based on a passing score of 60%. Given the positive skewness of the score distribution (*p* < 0.034), the median was used as the primary measure of central tendency to better reflect students’ performance. The distribution of the students’ scores is shown in Figure 1.

As the distribution of scores is skewed [Figure 1], a non-parametric procedure, such as the median method, is indicated in this case [34]. The overall median score of 18 out of 66 indicated that half of the students scored below 27% of the total possible score, revealing significant gaps in fundamental mathematical skills. With a passing threshold of 60% (40 points), the median score of 18.00 (27%) demonstrated that most students fell well short of the competency requirement.

A chi-square test revealed a significant difference in the proportion of male and female students scoring above the median (*X*^2^ = 4.34, *df* = 1, *p* = 0.037; see Table 1). Similarly, a Mann–Whitney U test confirmed a significant difference in knowledge scores between male and female students (*U* = 1423.5, *p* = 0.048). The bias-corrected and accelerated (BCa) 95% confidence intervals for the median scores further supported this difference, with male students scoring between 18.00 and 22.00, while female students consistently scored 17.00. Using the rank-biserial correlation, the effect size was small (*r* = 0.14), indicating a modest but meaningful difference.

We also examined the role of students’ level of study on their median test scores. A chi-square test was used to compare the proportions of students who scored above and at or below the median across the three levels of study. The results indicated that the proportions of students who scored higher than the median differ significantly across the three levels of study (see Table 2). Additionally, an effect size was calculated using Cramer’s V, which was found to be ≈0.217, indicating a small to medium effect size.

A Kruskal–Wallis test was conducted to determine whether there were significant differences in the median test scores across the three levels of study. This test compares the actual distributions of scores among groups rather than just proportions. The results indicated a significant difference in median scores among the groups; H (2) = 11.14; *p* = 0.004.

Post hoc pairwise comparisons were then performed. Statistically significant differences were found between first-year and third-year/seventh-semester students (Test statistic = −28.99; Standard Test Statistic = −3.153; *p* = 0.002; Adjusted *p* = 0.005) and between second-year and third-year/seventh-semester students (Test statistic = −25.170; Standard Test Statistic = −2.86; *p* = 0.004; Adjusted *p* = 0.013).

After applying the Bonferroni correction (adjusted significance threshold of α = 0.0167), these differences remained significant, indicating that third-year/seventh-semester students scored higher than both first- and second-year students. The difference between first-year and second-year students was not statistically significant (Test statistic = −3.82; Standard Test Statistic = −0.501; *p* = 0.616; Adjusted *p* = 1.00) (see Figure 2).

To shed more light on the students’ performance, the percentage of the total possible scores achieved and the interquartile range (IQR) of each component were compared (see Table 3). The IQR values for individual components ranged from 0 to 4, indicating varying levels of score dispersion across test sections. Notably, Component 4 had an IQR of 0, reflecting highly consistent performance at a low level, while Component 8 had the highest IQR (4), indicating a wider distribution of scores. Additionally, the IQR for total test scores across all components was 10 (Q1 = 14, Q3 = 24), meaning that the middle 50% of students scored between 14 and 24 out of the maximum possible 66 points. This suggests a moderate level of variability in overall student performance.

### 3.3. The Difficulty Level of Test Items

For a deeper understanding of their performance, Table 4 presents the mean *p* values of all the components based on students’ responses to each component item. The mean *p* values represent the average difficulty of items within each component, calculated as the sum of the *p* values for all items in a component divided by the number of items.

The analysis of item difficulty across components revealed variability in student performance. None of the items were easy. Most components had difficult items, with mean *p* values below 0.30, indicating consistent challenges for students. The second and fourth components consisted entirely of difficult items. In contrast, Components 5, 7, 8, 9, and 10 contained moderately difficult items. Overall, 25 (56%) items were difficult, while 20 (44%) were moderately difficult.

## 4. Discussion

The purpose of this study was to assess the basic mathematical skills of students enrolled in a Bachelor of Nursing program at a public university in Saudi Arabia and related factors. Our findings revealed that the vast majority of students were unable to pass the test. The low performance reflected in the median score raises real concerns about students’ ability to accurately perform dosage calculations, which are essential for safe medication administration. Inadequate mathematical competency can lead to medication errors, impacting patient safety not only in Saudi Arabia but globally. In practical terms, several barriers prevented the direct assessment of students’ ability to perform dosage calculations in clinical practice. The small-sized, non-teaching, and overcrowded local public hospital provides limited opportunities for students to prepare and administer medications under supervision. Research from the UK, Turkey, and Italy has already demonstrated that poor numeracy skills are a persistent concern. Given that similar trends are observed in other countries, addressing these gaps in mathematical education is a priority for nursing programs worldwide. In addition, the median score falling well below the 60% benchmark reflects a critical gap between students’ current mathematical abilities and the nursing program’s expectations for safe clinical practice. The results highlighted the need for targeted educational interventions, such as remedial courses or focused tutoring, to strengthen students’ mathematical skills. These interventions should be implemented during the first semester of the nursing program, before students begin core nursing courses in the second year. Additionally, reinforcing numerical competency within theoretical courses, such as ‘Introduction to Clinical Skills in Patient Care’, could help bridge the gap between mathematical knowledge and its application in nursing contexts. Ensuring that students acquire these competencies before clinical placements is crucial to minimizing medication errors and enhancing patient safety.

The students consistently struggled across multiple components (e.g., Components 1 to 4). Additionally, components with greater variability (Components 6 to 8) suggest varying levels of understanding among students. This outcome is not surprising, as it can be traced back to students’ formative school years, during which many did not acquire essential mathematical skills. This is perhaps due to the fact that school improvement relies heavily on principals and teacher leaders; yet these roles in Saudi Arabia are underdeveloped and lack sufficient support [40]. An international survey assessing key competencies in mathematics showed that 15-year-old Saudi Arabian students scored 373, which was considerably lower than the average score of 489 [41]. As a result, many students entered higher education with inadequate numerical proficiency [28]. Despite some efforts since the aforementioned survey, progress has been minimal, with Saudi Arabian students achieving a score of 389 in the most recent assessment, still falling short of the international average score of 472 [42].

The findings of this study are particularly concerning due to the high percentage of students (98%) who failed the test—far exceeding the failure rates reported in other countries. The findings of this study are important to readers in the Arab world and beyond, as they highlight a critical gap in mathematical competency among nursing students. Since dosage calculation skills are essential for patient safety worldwide, the results offer valuable insights for nursing educators and policymakers in other regions as well. Studies from the UK [4], Turkey [24], and Italy [3] reported that poor mathematics skills affected 55%, 52%, and 30% of nursing students, respectively. Given that mathematical deficits among nursing students are a global concern, with pass rates ranging from 45% to 70%, the 2% pass rate observed in this study underscores an especially severe gap in numeracy skills, necessitating urgent action.

In this study, students achieved a median score of 18 (27%). In comparison, second-year diploma students [4] and those of a Bachelor of nursing education program [24] achieved a median of 56.8% and 50%, respectively. Since medication dosage errors are unacceptable in practice [43], these results are concerning, particularly as third-year students are expected to answer most questions correctly [3].

Another possible factor contributing to the poor results in this study is that English is a second language for the students. While difficult or unfamiliar words were translated upon request, some students may have hesitated to ask for clarification. Pearson, a global learning company, states in a technical report that language proficiency can impact test performance in non-native speakers [44].

That said, English is the language of instruction in the nursing program, and 34 (76%) of the questions were short, using simple words or containing no wording at all. Therefore, while a language barrier may have played a minor role, its overall impact on performance is uncertain.

In addition, a similar minimal influence could have resulted from the testing environment (no calculators). The decision to prohibit calculator use was based on multiple factors: the test questions were designed to ensure that calculator use was unnecessary; the goal was to assess students’ fundamental mathematical skills [see Section 2.5]; the approach aligns with most relevant studies [3,24,28]; and potential disadvantages of calculator use were considered [4,45]. While calculators can help reduce computational errors [4], their use would have had limited utility—only aiding in a few questions—while also acting as a substitute for essential mathematical knowledge and masking students’ inability to perform basic calculations. As a result, any grade inflation would have been minimal, and the failure rate would still be at least 95%.

In this study, male students outperformed female students, though the effect size was small. This aligns with prior research suggesting that gender differences in math performance may stem from confidence [46] and anxiety [23]. However, the findings are mixed. Some studies attribute male students’ better performance to greater confidence [46] or early education trends, with boys outperforming girls in fourth grade [47], and as cited elsewhere [48]. Other researchers [3] also identified gender as a key factor in math proficiency. In contrast, a meta-analysis of 242 studies [49] concluded that math performance is similar across genders. Others [50] suggest that gender stereotypes held by female students and their families may negatively impact learning and performance.

The results of this study also revealed that third-year/seventh-semester students performed significantly better than first and second-year students. This difference may be attributed to senior students having taken nursing courses that involve mathematical calculations. Supporting this finding, other researchers [4] reported that older participants (aged 35 and above) were statistically significantly better at performing basic numerical calculations compared to younger participants. Similarly, students with very low math scores were typically at the junior level [51].

Previous studies on nursing students’ mathematical skills have overlooked the aspect of item difficulty. The analysis of the *p* value of the test’s items confirms the relationship between the difficulty index and the achievement of learning outcomes [52]. In the present study, it was carried out as an additional measure to verify the students’ knowledge gaps highlighted above. Addressing item difficulty might enhance educational practices in nursing programs, where educators can better identify areas where students may need additional support.

Students’ struggles with various components have important practical implications for educators. Given the consistent difficulties observed in Components 1 to 4, targeted instructional interventions should focus on these areas. These could include remedial workshops, personalized tutoring, or practice modules embedded in the curriculum. Virtual platforms like safeMedicate and eDose^TM^—which have been shown to be effective in teaching and assessing numeracy among undergraduate nursing students [3]—can also enhance students’ engagement and performance. Additionally, peer-assisted learning strategies could further support students struggling with these foundational topics. In contrast, the variability in Components 6 to 8 highlights the need for differentiated instruction to support struggling students while challenging those who excel. Component 5, which reflected moderate success, may not require immediate intervention but could benefit from periodic reinforcement to sustain performance.

## 5. Implications and Recommendations for Further Research

Although statistically significant differences in median scores were observed based on gender and level of study, these findings lose practical relevance in light of the overwhelming 98% failure rate. The near-universal failure suggests that fundamental mathematical deficits are pervasive across student groups, irrespective of gender or academic level. This reflects a systemic issue that requires broader interventions beyond targeting specific subgroups.

The implications for nursing education are alarming, as mathematical competence is essential for safe clinical practice and for public protection. Accurate dosage calculations and medication administration are critical to ensuring patient safety. However, since mathematics is not currently part of the nursing curriculum, innovative strategies must be implemented to develop foundational numeracy skills throughout the program. The findings also raise concerns about the adequacy of admission criteria and curriculum design, as students at advanced stages of the program were unable to demonstrate the required mathematical competence.

Addressing these deficiencies also requires targeted reforms in public school education to ensure students graduate with adequate mathematical proficiency—an essential skill for both professional success and societal productivity [41]. This call for reforms is not new per se and was recently issued by the Organization for Economic Co-operation and Development (OECD) [40]. Graphic calculators first appeared in the mid-1980s [53] and introduced many new and exciting changes to mathematics education [54]. Research has shown that these calculators are useful in mathematics teaching and learning [55,56] for students at the age of 15 years and beyond.

In advancing nursing education and ensuring the development of highly qualified graduates in Saudi Arabia, policymakers may look to examples from countries like Malta [57] and the United States, where many nursing schools require passing the TEAS for program admission [58]. However, implementing a similar approach in Saudi Arabia could present significant challenges due to differences in the nursing education systems, the lack of existing entrance exams, potential political and administrative constraints, and possible limitations in institutional support.

As an alternative, universities could assess and address students’ mathematical competency through methods reported earlier in the ‘Discussion’ section. These interventions could identify gaps early on and allow for targeted remediation without the need for a full entrance exam. While such measures may require significant changes in the current curriculum, they offer a more feasible solution for strengthening mathematical skills prior to students engaging in more complex nursing coursework.

Enacting the recommendations outlined here is a step in the right direction that policymakers must take into account to ensure the safety of patients and public trust in healthcare professionals. Ensuring nursing students are equipped with strong numeracy skills is not merely an academic requirement but a cornerstone of public safety.

Further research across different regions of Saudi Arabia would help identify areas where educational reforms are most needed. Additionally, multi-site studies in nursing programs across the Arab world, including Saudi Arabia, would provide valuable insights into the broader issue of mathematical proficiency among nursing students. Such studies would improve sample representativeness and explore the potential benefits of integrating clinical simulation-based dosage calculation assessment. To account for the potential language barrier, future research should include a preliminary English proficiency assessment and consider administering bilingual test versions to determine the extent of this effect. Future studies should compare manual vs. calculator performance to refine nursing education strategies.

Future research should also incorporate a standardized measure of question difficulty to provide a more nuanced understanding of students’ mathematical skills. Developing a framework for categorizing question difficulty—considering cognitive demand, question type, and performance data—would ensure assessments are balanced and aligned with intended learning outcomes. This approach would help educators distinguish between performance issues due to limited mathematical knowledge and those stemming from the inherent complexity of the test items.

In cultures where mathematics education relies primarily on rote memorization rather than conceptual understanding, students may face challenges in nursing education [59]. Future studies should incorporate a diagnostic tool to assess students’ learning approaches, providing deeper insight into their mathematical performance. Additionally, differences in educational backgrounds continue to impact mathematics proficiency [59]. Future research could address this by including measures of students’ prior mathematics education.

In summary, while gender- and level-specific differences offer interesting statistical insights, the near-total failure rate underscores the urgent need for institution-wide interventions and curricular reforms. These changes are crucial for equipping students with the numeracy skills required for clinical practice and safeguarding both patient safety and public protection.

## 6. Study Limitations

This study has its limitations. Since it was conducted at a single university branch, caution is needed when generalizing the findings to other nursing programs within Saudi Arabia or internationally. Additionally, the small sample size limits the robustness of the findings. A retrospective power analysis indicated that the achieved power (0.43) falls below the desired threshold of 0.80, which is required to detect a meaningful effect. Moreover, the response rate (40.6%) introduces a highly unlikely risk of selection bias, as the characteristics of non-respondents may differ from those who participated, potentially skewing the results. However, this risk is likely low. Another limitation was that this study did not include an objective measure of English proficiency. Although translation assistance was provided, language barriers may have influenced some students’ performance.

This study did not include a survey or diagnostic tool to assess whether students relied on rote memorization or conceptual understanding, which could have provided deeper insight into their poor performance on the mathematics test. Additionally, the absence of a pre-test questionnaire assessing students’ prior mathematics coursework made it difficult to determine the extent to which educational background influenced mathematical performance. Finally, this study did not directly assess students’ ability to perform dosage calculations in clinical practice, which would have provided a real-world measure of competency.

## 7. Conclusions

Despite its limitations, this is the first study in the Arab world to examine nursing students’ basic mathematics skills. This study highlights a severe deficiency in mathematical skills among nursing students, exceeding gaps previously reported in international research. It is essential that nursing education programs take responsibility for ensuring that graduates possess the mathematical competencies required for accurate medication dosage calculations. Addressing these deficiencies is vital to enhancing patient safety and strengthening nursing education worldwide.

## Figures and Tables

**Figure 1 nursrep-15-00150-f001:**
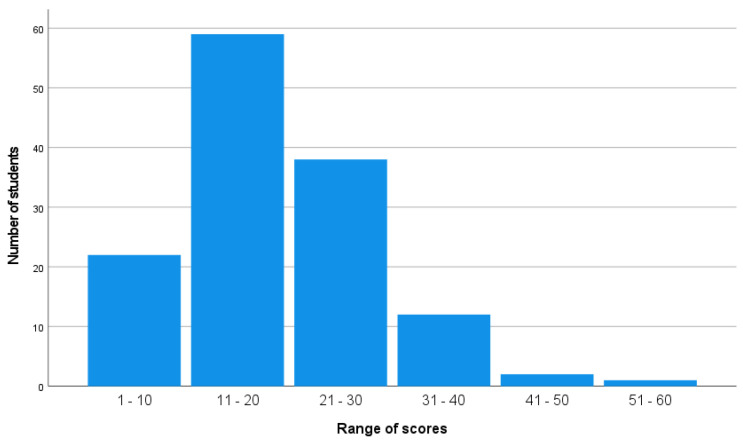
Distribution of scores.

**Figure 2 nursrep-15-00150-f002:**
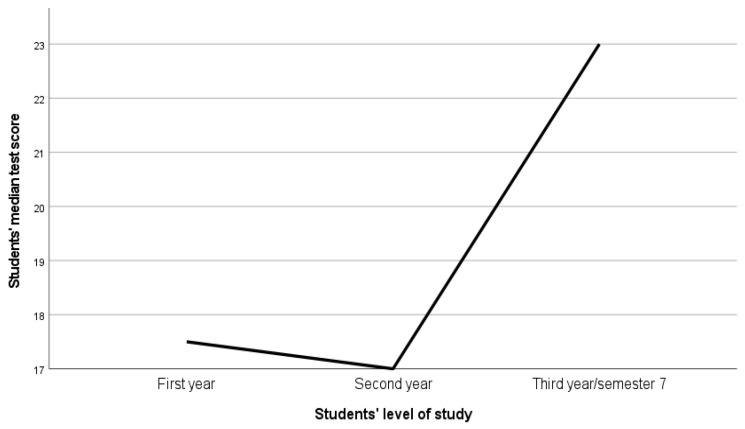
Relationship between students’ median test scores and level of study.

**Table 1 nursrep-15-00150-t001:** The chi-squared test of students’ median knowledge scores based on gender.

	Male Students	Female Students	Total
>Median	52 (54%)	13 (34%)	65 (42%)
≤Median	44 (46%)	25 (66%)	69 (58%)
Total	96	38	134 (100%)

*X*^2^ = 4.34; *df* = 1; *p* = 0.037.

**Table 2 nursrep-15-00150-t002:** The chi-squared test (all expected frequencies are greater than 5) of the students’ median knowledge scores based on the level of study.

	1st Year	2nd Year	3rd Year/7th Sem.	Total
>Median	22 (48%)	21 (36%)	22 (76%)	65 (49%)
≤Median	24 (52%)	38 (64%)	7 (24%)	69 (51%)
Total	46	59	29	134 (100%)

*X*^2^ = 12.64; *df* = 2; *p* = 0.002.

**Table 3 nursrep-15-00150-t003:** Percentage of total possible scores achieved and the IQR across test com.

	% of Total Possible Scores Achieved	IQR		% of Total Possible Scores Achieved	IQR
Component 1	13.6%	1	Component 6	23%	4
Component 2	16%	1	Component 7	32%	2
Component 3	27%	1	Component 8	38%	4
Component 4	11%	0	Component 9	30%	2
Component 5	50%	1	Component 10	39%	3

**Table 4 nursrep-15-00150-t004:** Components’ level of difficulty.

	Range of *p* Values	Mean *p* Value
Component 1	0.05–0.38	0.14
Component 2	0.11–0.22	0.17
Component 3	0.04–0.60	0.23
Component 4	0.02–0.16	0.11
Component 5	0.30–0.80	0.55
Component 6	010–0.34	0.23
Component 7	0.22–0.51	0.33
Component 8	0.20–0.51	0.39
Component 9	0.19–0.49	0.34
Component 10	0.20–0.49	0.37

Note: Based on Mitra et al. [39], items with *p* values > 0.80 are easy; *p* values between 0.30–0.80 are moderately difficult; and *p* values < 0.30 are difficult.

## Data Availability

The raw data supporting the conclusions of this article will be made available by the author on request.

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
