# Peer review of "An Indispensable Requirement for Medical Dosage Calculation: Basic Mathematical Skills of Baccalaureate Nursing Students"

_nursrep, 2025, doi:10.3390/nursrep15050150_

Round 1

Reviewer 1 Report

Comments and Suggestions for Authors

This is an interesting paper that highlights the lack of math competences in nursing students and how it could affect medication administration and patient safety. Nonetheless, I think that before publishing some improvements to the manuscript should be made.

Materials and methods:

Sample size calculation and sampling method must be stated. Provide a brief description on how the sampling was done.

Item analysis: section 2.8. I suggest a brief description on how it was done

section 2.9 lines 180-181 some relevant information appears to be missing. 

Line 183. Before pairwise comparisons, a test must be done to compare all groups simultaneosly, i.e. the Kruskal-Wallis test, and if this last one is significant, then a pair-wise test must be done as a post-hoc test.

Results. Section 3.2

It is not clear if the used parametric or non-parametric test. Although they said data were not normally distributed, some times they refer means, standar errors and standard deviations (which are used when data are normally distributed). 

Table 3. What does the superscript refers in SD2?

Figure 2. Check y-axis title. It says "mean" instead of "median"

Discussion: Lines 289-291 are repeated. 

Comments on the Quality of English Language

English language. The english must be carefully checked. Use past tense. Check grammar i.e. lines 161-162.

Author Response

Comment 1: Sample size calculation and sampling method must be stated. Provide a brief description on how the sampling was done.

Response 1: I've responded to this comment. Please see page 3, lines 97-100.

Comment 2: Item analysis: section 2.8. I suggest a brief description on how it was done.

Response 2: My response is presented on page 5, lines 184-186.

Comment 3: Section 2.9 lines 180-181 some relevant information appears to be missing. 

Response 3: This has been revised. Please see page 5, lines 191-195.

Comment 4: Line 183. Before pairwise comparisons, a test must be done to compare all groups simultaneously, i.e. the Kruskal-Wallis test, and if this last one is significant, then a pair-wise test must be done as a post-hoc test.

Response 4: This comment was addressed. Please see 5, lines 196-201.

Comment 5: It is not clear if the used parametric or non-parametric test. Although they said data were not normally distributed, sometimes they refer means, standard errors and standard deviations (which are used when data are normally distributed). 

Response 5: Immediately below Figure 1, I mentioned non-parametric procedure. This and all other concerns were addressed. Please see page 5 line 217. The means, standard errors, and standard deviations were removed. 

Comment 6: Table 3. What does the superscript refers in SD2?

Response 6: The Table was revised and SD was removed. The superscript is a footnote. 

Comment 7: Figure 2. Check y-axis title. It says "mean" instead of "median". This was corrected. Please see the bottom of page 7.

Comment 8: Lines 289-291 are repeated. 

Response 8: The repeated content has been removed. Please see page 10, lines 335-337.

Comment 9: The English must be carefully checked. Use past tense. 

Response 9: I addressed this issue to the best of my knowledge. If more work needs to be done, I can consult a language editing service. 

Comment 10: Check grammar i.e. lines 161-162.

Response 10: This was corrected. Please see page 4, line 168. 

Reviewer 2 Report

Comments and Suggestions for Authors

The study does not show participants' background on prior mathematical knowledge. This makes assessing for context in which participants have no prior knowledge unjustifiable. It would be appropriate if a prior level of mathematical background is acknowledged, further assessing the need for improvement after identified gaps. 

Author Response

Comment 1: The study does not show participants' background on prior mathematical knowledge. This makes assessing for context in which participants have no prior knowledge unjustifiable. It would be appropriate if a prior level of mathematical background is acknowledged, further assessing the need for improvement after identified gaps. 

Response 1: I appreciate the reviewer’s comment regarding students’ prior knowledge of mathematics. This study assessed nursing students' possession of basic mathematics skills. All nursing students, worldwide, whose mathematics skills were evaluated in previous publications learned mathematics at schools before proceeding to university. My question is: which nursing students without prior mathematics knowledge should have their basic mathematics skills assessed? Our findings highlight a crucial issue in nursing education: mathematical incompetency increases the risk of committing medical dosage calculation errors. Addressing this gap is essential for improving patient safety.

Additionally, prior studies assessing nursing students’ mathematical competence were considered justified, as evidenced by their publication in high-quality, peer-reviewed journals. Each of these studies was conducted in a single country. Why, then, is our study not justified, especially when it is the first to be reported from the Arab world which is a large region? A critical gap exists in the literature regarding this issue in Arab countries, and our study provides essential insights into this largely unexplored context.

Comment 2: The study can also recommend either an entry-level mathematical background or basic mathematical course as an additional module.

Response 2: I appreciate the reviewer’s suggestion regarding recommendations for nursing education. While our study highlighted the concerning deficiencies in students’ basic mathematical competence, the issue extends beyond nursing education and requires earlier intervention at school level before they enroll higher education. However, we acknowledge that nursing programs could consider assessing entry-level mathematical proficiency or integrating a basic mathematics course as an additional module to support students who struggle with fundamental calculations. This suggestion aligns with our findings and has been incorporated into the 'Discussion' section (please see the pages 9-10, lines 315-320.

Comment 3: The study does not show participants' background on prior mathematical knowledge. This makes assessing for context in which participants have no prior knowledge unjustifiable. It would be appropriate if a prior level of mathematical background is acknowledged, further assessing the need for improvement after identified gaps.

Response 3: While the study did not include a pre-test questionnaire on participants’ mathematics background, this limitation has been acknowledged in the revised 'Study Limitations' section (please see page 13, lines 488-491. 

Based on the first comment above, one concludes that assessing a student in a context that s(h)e has prior knowledge cannot be justified. Based on the present comment, one concludes that assessing a student in a context in which s(h)e had no prior knowledge cannot be justified. It appears that both comments are contradictory. 

Reviewer 3 Report

Comments and Suggestions for Authors

The study addresses a critical issue in nursing education-mathematical literacy and its impact on medication safety. The cross-sectional design, statistical analyses (Mann-Whitney U test, median test, difficulty index), and clear methodology enhance reliability. Educational and policy relevance: The study provides valuable recommendations for improving nursing curricula, particularly regarding remedial math courses and early intervention. The following recommendations are provided for reference

1.The study assumes that poor math skills lead to medication errors, but does not directly test students' ability to perform dosage calculations in clinical practice. (Line: 265-268). Suggested: Include a discussion of the lack of direct clinical validation and the need for clinical simulation-based dosage calculation assessments in future research. Alternatively, reference existing studies that demonstrate a direct relationship between math skills and actual medication errors.

2.While the study acknowledges its limitations (lines 392-396), it should also address: Selection bias due to the 40.6% response rate, which may skew the results. The need for multi-site studies across Saudi Arabia to improve sample representativeness. Suggested adding a brief discussion of selection bias and explicitly recommending future multi-site research.

3.The paragraph suggesting that English as a second language may have affected test performance needs further development: (Line: 307-310). This argument is largely speculative and would benefit from supporting evidence. Consider including questionnaire data or interviews with students to verify that language barriers actually prevented students from asking questions. Avoid unfounded gender assumptions: Suggesting that "particularly female students" may have been too shy to ask questions lacks empirical support and risks perpetuating gender stereotypes.Either provide literature support for gendered communication patterns in Saudi Arabian educational contexts or adopt more neutral language.

4.If language proficiency is considered a significant variable, future studies should include a preliminary assessment of English language proficiency. Consider conducting a comparative study with test versions in both English and Arabic. Provide a more detailed description of the translation assistance protocol in the methods section. This language barrier hypothesis should be explicitly addressed in the study limitations section. Outline specific approaches for controlling this variable in future research.

5.The study does not discuss the impact of calculator use, despite its relevance to real-world nursing practice. The test environment (no calculators) may not accurately reflect clinical conditions and may overstate the failure rate (98%). Suggested adding a discussion of how calculator limitations may affect test results (lines 310-312 or 352-355). Consider suggesting future studies comparing manual vs. calculator performance to refine nursing education strategies.

6.The study suggests remedial courses but does not specify how these should be implemented in the nursing curriculum (line 336-338). More details are needed on whether these interventions should be offered as prerequisite courses before entering the nursing program, integrated into existing courses such as pharmacology, or required as a competency assessment prior to clinical training. Providing these details would strengthen the practical implications of the study.

7.The feasibility of implementing TEAS entrance exams in Saudi Arabia is unclear due to differences in the nursing education system (line 366-370). The study should discuss potential challenges (e.g., lack of existing entrance exams, political constraints) and suggest alternative solutions, such as pre-nursing math courses or first-semester math proficiency tests with remediation. This would make the recommendations more practical and applicable.

8.The study does not adequately emphasize its limitations, particularly with regard to cultural factors, prior educational disparities, and lack of direct clinical assessment.

Author Response

Comment 1: The study assumes that poor math skills lead to medication errors, but does not directly test students' ability to perform dosage calculations in clinical practice. (Line: 265-268). Suggested: Include a discussion of the lack of direct clinical validation and the need for clinical simulation-based dosage calculation assessments in future research.

Response 1: Addressed as suggested. Please see page 9, lines 304-308 and page 12, lines 447-448. In addition, lack of direct clinical validation was acknowledge as a limitation. Please see page 13, lines 489-491.

Comment 2: While the study acknowledges its limitations (lines 392-396), it should also address: Selection bias due to the 40.6% response rate, which may skew the results. The need for multi-site studies across Saudi Arabia to improve sample representativeness. Suggested adding a brief discussion of selection bias and explicitly recommending future multi-site research.

Response 2: I appreciate the reviewer's suggestions. The comment on 'selection bias' was taken into account. More information can be seen on page 13, lines 48-480. With regards to 'multi-site studies', I've already highlighted the importance of larger-scale studies in nursing programs across Saudi Arabia and the Arab world (Page 11, lines 377-379). My recommendation has now been refined (Page 12, lines 445-448). 

Comment 3: The paragraph suggesting that English as a second language may have affected test performance needs further development: (Line: 307-310). This argument is largely speculative and would benefit from supporting evidence. Consider including questionnaire data or interviews with students to verify that language barriers actually prevented students from asking questions. Avoid unfounded gender assumptions: Suggesting that "particularly female students" may have been too shy to ask questions lacks empirical support and risks perpetuating gender stereotypes. Either provide literature support for gendered communication patterns in Saudi Arabian educational contexts or adopt more neutral language.

Response 3: I appreciate the reviewer’s suggestion to verify whether language barriers prevented students from asking questions. However, I’d like to clarify that there was no language barrier between the students, the investigator, and his colleague as all share the same mother tongue. The concern was the students' command of English as a second language could have limited their ability to fully understand some of the test questions. To address this, and based on student requests, the investigator and the nursing colleague on the female campus provided Arabic translations for any words or phrases that students found difficult to understand. Because the study did not assess language proficiency among students, this was acknowledged as a limitation. In addition, gender assumptions were avoided along with maintaining a neutral language (Page 10, lines 352-359).

 Comment 4: If language proficiency is considered a significant variable, future studies should include a preliminary assessment of English language proficiency. Consider conducting a comparative study with test versions in both English and Arabic. Provide a more detailed description of the translation assistance protocol in the methods section. This language barrier hypothesis should be explicitly addressed in the study limitations section. Outline specific approaches for controlling this variable in future research.

Response 4: The potential influence of language proficiency on students' performance has been discussed on page 10, lines 352-355 and 356-359. In addition, recommendations for future research to include preliminary assessment of English command and to conduct comparative study using Arabic and English version of the test were presented on page 12, lines 450-453. In section 2.7 (Data Collection Process), I indicated that no translation assistance protocol was used (page 4, lines 173-174). I also addressed the language barrier issue in the study limitations section (page 13, lines 483-485). Concerning specific approaches to control for language barrier in future research, please see page 12, lines 450-453.

Comment 5: The study does not discuss the impact of calculator use, despite its relevance to real-world nursing practice. The test environment (no calculators) may not accurately reflect clinical conditions and may overstate the failure rate (98%). Suggested adding a discussion of how calculator limitations may affect test results (lines 310-312 or 352-355). Consider suggesting future studies comparing manual vs. calculator performance to refine nursing education strategies.

Response 5: I have already discussed the rationale for prohibiting calculator use and its effect on test results on pages 10-11, lines 360-369. I explained the reasoning behind this decision, its alignment with relevant studies, and its expected minimal impact on the failure rate. Given this, I believe the concern has been adequately addressed. In 'Implications and Recommendations for Further Research' section, I suggested that future studies should compare manual vs. calculator performance to refine nursing education strategies (page 12, lines 453-454).

Comment 6: The study suggests remedial courses but does not specify how these should be implemented in the nursing curriculum (line 336-338). More details are needed on whether these interventions should be offered as prerequisite courses before entering the nursing program, integrated into existing courses such as pharmacology, or required as a competency assessment prior to clinical training. Providing these details would strengthen the practical implications of the study.

Response 6: The pages 9-10, lines 315-320 address the above comment. 

Comment 7: The feasibility of implementing TEAS entrance exams in Saudi Arabia is unclear due to differences in the nursing education system (line 366-370). The study should discuss potential challenges (e.g., lack of existing entrance exams, political constraints) and suggest alternative solutions, such as pre-nursing math courses or first-semester math proficiency tests with remediation. This would make the recommendations more practical and applicable. 

Response 7: Page 12, lines 429-440 address the above comment.  

Comment 8: The study does not adequately emphasize its limitations, particularly with regard to cultural factors, prior educational disparities, and lack of direct clinical assessment.

Response 8: I appreciate the reviewer’s insightful comments regarding the study’s limitations. In response, I have expanded the 'Study Limitations' section to acknowledge the potential influence of cultural factors and prior educational disparities on students’ mathematical performance (page 13, lines 486-491). While these factors are beyond the direct control of this study, future research could explore their impact through targeted surveys or pre-test assessments. Additionally, I have clarified the absence of direct clinical assessment of dosage calculations as a limitation (page 13, lines 491-493). 

Round 2

Reviewer 2 Report

Comments and Suggestions for Authors

Accepted

Reviewer 3 Report

Comments and Suggestions for Authors

While I acknowledge the relationship between nursing students' mathematical skills and their impact on medication dosage calculations and patient safety, I remain somewhat cautious about the extent of this relationship given that many medical orders are now computerized and automated. Nevertheless, it's clear that the author has made significant revisions to the manuscript. As a topic in nursing research, this remains a topic worthy of attention. I believe that this revised paper is of good quality and suitable for publication.